# Echoes of the Past: Unveiling the Kharga Oasis' Cultural Heritage and Climate Vulnerability through Millennia

Hossam Ismael [1,2,*], Waleed Abbas [3], Heba Ghaly [4] and Ahmed M. El Kenawy [5]

1  Department of Geography & GIS, Faculty of Arts, New Valley University, Kharga 72511, Egypt
2  British University in Egypt, Cairo 11837, Egypt
3  Department of Geography & GIS, Faculty of Arts, Ain Shams University, Cairo 11566, Egypt; walid.abbas@art.asu.edu.eg
4  Department of Archeology, Faculty of Arts, Helwan University, Cairo 11795, Egypt; heba_mohammed@arts.helwan.edu.eg
5  Instituto Pirenaico de Ecología, CSIC, Campus de Aula Dei, 1005, 50059 Zaragoza, Spain; kenawy@mans.edu.eg
*  Correspondence: hossam.ismael@art.nvu.edu.eg or hossam.ismael@bue.edu.eg

**Abstract:** The civilization and tangible cultural heritage of the Kharga Oasis has a historical precedence over that of the old Nile Valley civilization. Approximately 12,000 years ago, a significant prehistoric migration occurred from the Kharga Oasis to the Nile Valley. This event was motivated by climate change and the southward shift of the Inter-Tropical Convergence Zone (ITCZ), which caused a shift in Egypt's savannah forests from abundant vegetation to an extremely dry desert. The present study investigates the progressive deterioration of the tangible cultural and civilized legacy of the Kharga Oasis over the course of several millennia, positing that this phenomenon can be attributed to the area's vulnerability to paleoclimatic fluctuations. The evaluation of the Kharga Oasis' susceptibility to climate change was predicated on the scrutiny of petroglyphs that were unearthed at different sites within the Oasis. This analysis was reinforced by paleoclimate information and radiocarbon dating (C14). The utilization of an interdisciplinary approach yielded significant insights into the dynamic climate patterns and their effects on the Kharga Oasis across temporal scales. The results illustrated a noteworthy alteration in climate, which caused the conversion of the Oasis terrain from being heavily wooded to becoming arid, mainly due to extended periods of drought. The present research postulates a novel and alternate hypothesis concerning the archaeological chronology of human habitation in the Kharga Oasis from ancient eras, predicated on the analysis of pictorial depictions on rock surfaces. The findings of this study made a noteworthy contribution to the current corpus of knowledge regarding the vulnerability of the ancient Egyptian society to the impacts of climate variability. Moreover, the petroglyphs' depictions provided a distinctive viewpoint on the climatic fluctuations that occurred in the Sahara and North Africa throughout the Holocene epoch, as well as the fundamental causative factors.

**Keywords:** paleoclimate change; vulnerability; ITCZ; civilized heritage

## 1. Introduction

The Western Desert of Egypt, including its Oasis, holds valuable records of ancient climatic events and the resulting economic and social transformations that have influenced the population from prehistoric times to the present. The Kharga Oasis, in particular, has witnessed significant climatic changes over the past 12,000 years, which have been extensively studied and documented using evidence from the Kharga-Dakhla Oases [1,2].

The Kharga Depression boasts a significant historical background and harbors archaeological sites and vestiges of past civilizations. The region has been populated since ancient times, as archaeological findings indicate human habitation spanning several millennia. The oases located within the depression have played a significant role as pivotal trading and

cultural hubs along the historic trade routes that traverse the Western Desert. Dating back 12,000 years, the Kharga Oasis provides abundant evidence of human settlements [3–5]. The Kharga Oasis has experienced a substantial concentration of population and commercial activity since 10,500 B.C., continuing into the Pharaonic period [6–8]. However, the northern parts of the Kharga Oasis were likely completely abandoned due to paleoclimatic changes, though population declines may have occurred during certain periods. Climatic and archaeological evidence suggests that 12,000 years ago, the Oasis had a greener environment with higher rainfall and more moderate temperatures, supporting both human and animal populations. However, as a result of climate change, the Oasis has become one of the harshest regions on the planet [6,9–11].

Herodotus, a Greek historian of the 5th century B.C., famously referred to Egypt as "A gift of the Nile", highlighting the river's pivotal role in the development of the great Egyptian civilization. However, this gift was a consequence of climatic changes that occurred 12,000 years ago. The relocation of the Inter-Tropical Convergence Zone (ITCZ) 800 km to the south resulted in the transformation of the savannah forests in Egypt's Western Desert into an arid desert, compelling the population to migrate eastward in search of fertile soil and rivers [11–14]. Geological and archaeological evidence from the Egyptian Western Desert showcases the region's dramatic climatic variability, leading to a gradual drying over thousands of years. These conditions forced residents to relocate closer to water sources for survival, resulting in the migration of many ancient Egyptians to the Nile over the past 12,000 years [1,6,15].

All geomorphological and archaeological proof from the hyper-arid Egyptian Desert reveals significant climatic and environmental shifts over the past 12,000 years that frequently do not correspond to climate anomalies noted in high-latitude archives. Therefore, the Kharga Oasis's geomorphological and archaeological archives show a shift from semi-arid to semi-humid conditions in approximately 10,500 BP. This change led to monsoonal rainfall and the spread of wild fauna, prompting prehistoric populations to reoccupy the entire area. Over the next 3200 years, relatively stable, humid conditions prevailed [16,17]. The southward shift of the Inter-Tropical Convergence Zone (ITCZ) caused a shift in Egypt's savannah forests from abundant vegetation to an extremely dry desert at 7300 BP. This can be traced through the discontinuance of sedimentary records of aquatic deposits at decreasing latitudes. The geological archives and archaeological evidence suggest a gradual desiccation and environmental deterioration of the Eastern Sahara, despite transitory climatic perturbations common to desert margins. The discontinuance of sedimentary records at decreasing latitudes indicates a deterioration of the environment [18–21].

The archaeological sites in the Kharga Oasis hold exceptional significance for Egypt's national identity and future, representing ancient civilizations that have left behind enduring cultural practices, historical documentation, and imprints [22,23]. The region's dense population during the Holocene, supported by permanent groundwater activity, has resulted in an abundance of archaeological sites that testify to diverse cultural, social, and economic changes alongside robust paleoclimatic shifts. Numerous works, since the early 20th century, have addressed cultural and climatic sequences at the regional scale [21,24–27].

Archaeological field works identified four major prehistoric cultural stages in the Kharga Oasis based on C14 dating of prehistoric human tools. These stages are Incomplete Occupation 11.500; Migration 10.000–8.000; Occupation 8.000–6.000; and Re-Occupation 6.000–4.700 BP [1,15,28,29]. Archaeological excavations have uncovered pottery vessels of varying sizes and shapes, indicating distinct functions and time periods. In addition, the discovery of buried water channels connecting temples and residential communities to wells or fresh lakes sheds light on a climatic transition that peaked approximately 6000 years ago [30]. However, there is a lack of research on the effects of climate change on Egypt's cultural and civilized heritage, particularly concerning archaeological monuments. Unfortunately, few studies have examined the impact of paleoclimate on the region's cultural and civilized heritage.

The Intergovernmental Panel on Climate Change (IPCC) [31] defines the vulnerability of tangible cultural heritage to paleoclimatic changes as the susceptibility or effect of climate change hazards, such as climate variability and extreme weather events, on particular cultural and civilized values. Since the publication of the Sixth Assessment Report of the IPCC, monitoring the vulnerability of cultural heritage to climate change has received considerable attention from a variety of disciplines [31]. While vulnerability to paleoclimatic change has been extensively studied, the physical and cultural heritage aspects have received scant attention [23,32]. Although research in this field began in the early 1970s, there has been a surge in interest in recent years. In the midst of ancient and substantial climatic changes, periodic field studies have revealed the prevalence of cultural and civilizational artifacts. Nevertheless, there is an urgent need for investigations that address both the vulnerability to climate change and tangible cultural heritage.

This study presents a novel theory regarding human occupation and the archaeological sequence in the Kharga Oasis, focusing on the influence of paleoclimatic changes. This theory builds upon previous research findings and incorporates the analysis of rock art and insights recorded by prehistoric Egyptians regarding their lifestyle, economic activities, cultural practices, climatic conditions, and environmental changes during the Holocene. This approach allows to bridge the current research gap by collecting available paleoclimate data from the Middle Holocene in the Kharga Oasis, providing a detailed account of the temporal, social, and spatial context of regional cultural and civilized activities when the Oasis was generally habitable between 7500 and 12,000 BP [1,2,6,6,9,17,19,20,25,30,33–36].

To shed light on the susceptibility of the ancient Egyptian civilization to climate change, the current research analyzes rock art paintings discovered in the Kharga Oasis as physical evidence of ancient climatic changes. This innovative method offers a fresh perspective on the archaeological sequence of prehistoric human occupation in the Kharga Oasis. This study also investigates the causes of climate-induced changes from a holistic climatic perspective, incorporating the interpretation of rock art depictions of animals and landscapes to understand the vulnerability of tangible cultural heritage. Additionally, field measurements and laboratory analyses are conducted on a subset of the Kharga Oasis archaeological sites to assess the vulnerability of tangible cultural heritage to recent climate change.

The primary objective of this study is to provide a comprehensive understanding of the paleoclimatic characteristics of the Kharga Oasis, Egypt. Our specific aims are to analyze the evidence of past climate variations as depicted in prehistoric rock art, investigating the potential of such art to determine the vulnerability of archaeological sites in the Kharga Oasis to historical climatic changes. Our research also seeks to understand the factors that caused climatic fluctuations in the Kharga Oasis and the possible impacts of these paleoclimatic changes on ancient civilizations in the Oasis. The current work relies on a comprehensive examination of multiple rock art sites, archaeological inscriptions, and scribbles located in the Kharga Oasis to facilitate a more profound understanding of the climatic and environmental changes that were encountered by the ancient Egyptians. The rock art records hold significant value as sources of information, offering insights into historical landscapes and enabling comprehension of the complex interplay among climate change, geography, and the safeguarding of the cultural heritage of ancient Egypt.

## 2. Materials and Methods

In order to examine the susceptibility of the tangible cultural and civilized heritage in the Kharga Oasis to the effects of climate change, the present study utilized the following research methodologies.

### 2.1. Study Area

The Kharga Oasis is situated between longitudes 30°27″ and 30°47″ E and latitudes 22°30′16″ and 26°00′00″ N (Figure 1). It is positioned at approximately 560 km towards the southwest of Cairo. The depression encompasses a landmass of approximately

22,000 square kilometers and is encompassed by parched desert terrains. The Kharga Depression constitutes a prominent topographical characteristic situated in the western region of Egypt [22,37]. From a geological perspective, it is distinguished by extensive expanses of sand dunes and rocky plateaus. According to the Koppen climate classification, the Kharga Oasis has an extreme desert climate type' (BWh) climate [38]. The Oasis plays a crucial role as an essential water reservoir within the severe and parched desert setting.

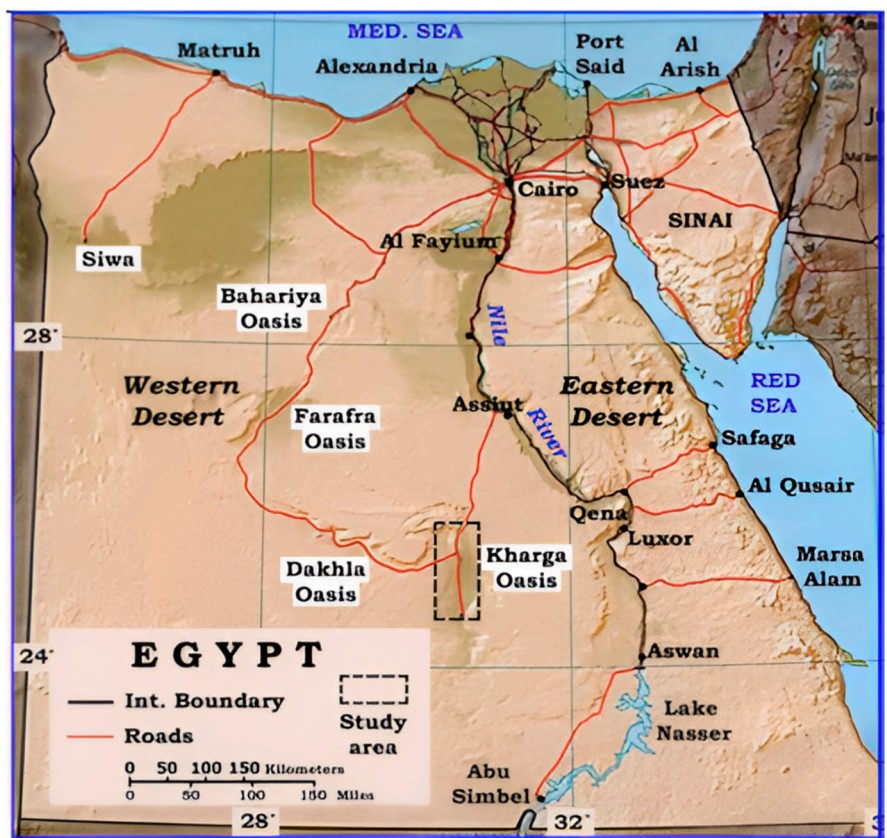

**Figure 1.** Location of the Kharga Oasis.

The hydrological processes of the Nubian sandstone formations facilitate the sustenance of vegetation and agricultural practices in the Oasis through the natural drainage of groundwater. Geological evidence suggests that the Kharga Oasis has long served as a habitat for Nile Valley settlers and communities. The natural drainage of groundwater from the Nubian sandstone into the Kharga soils made it possible for prehistoric populations to congregate, inhabit, and cultivate the famed oases within the vast The Kharga-Dakhla depression [37]. Specifically, the Kharga has a completely unique identity; it is considered an ideal Oasis, a shelter, a station for passing nomads, in which numerous cultural and civilized effects are recomposed.

### 2.2. Literature Review

The present study conducted an extensive literature review to gather information on the current state of knowledge regarding the impacts of climate change on cultural heritage, as well as previous research conducted in analogous contexts. This facilitated the establishment of a theoretical framework for this study and enabled the identification of areas where further research is needed. A thorough analysis of diverse climatic and geographic studies has been conducted to investigate the susceptibility of the ancient Egyptian civilization to the effects of climate change. The objective of this review was to ascertain the primary factors that exert an influence on the vulnerability of archaeological sites to impacts related to climate. This study delved into several significant factors,

including migration causes, social dynamics in ancient Egyptian society, cultural heritage, drought, alterations in pottery styles, and shifts in agricultural practices.

In the second half of the 20th century, several scientific fields work focused on Kharga's prehistory, including the Combined Prehistoric Expedition (CPE), the University of Kansas Western Desert Expedition, the Joint Cairo University/Egyptian Geological Survey Project, and the Kharga Oasis Prehistory Project. Between 1981 and 1987, Siegbert Eickelkamp, an engineer with the Abu Tartur phosphate mines, spotted 136 prehistoric sites on the northwestern margin of the depression. Since 2001, the North Kharga Oasis Survey has discovered several prehistoric and early dynastic petroglyph panels [28,34,39–41]. However, studies related to the Kharga's prehistory were still at an early stage compared to the rest of the Western Desert. Recent research by the Institut français d'archéologie orientale (IFAO) in Kharga has collected new evidence, allowing the proposal of a renewed archaeological sequence for human occupation in the Kharga Oasis [40,42,43].

The present study involved a comprehensive examination of multiple rock art sites, archaeological inscriptions, and scribbles discovered within the Kharga Oasis. In this work, a meticulous selection process was implemented, which entailed evaluating more than 50 articles. This review illuminates the intricate interdependence between climatic and geographic variables, revealing the multifaceted dynamics that influenced the development of historical civilization. The examination of migration factors sheds light on the adaptive strategies employed by past societies in response to shifting environmental conditions. Similarly, the analysis of social structures in ancient Egypt offers valuable perspectives on the ways in which societal dynamics intersected with the changing climate. In addition, the investigation of cultural heritage, encompassing the scrutiny of pottery styles and other archaeological remnants, reveals the impact of climate change on the perpetuation and persistence of historical customs, ceremonies, and convictions.

### 2.3. Paleoclimate Derived Data

The Kharga Oasis possesses considerable archaeological wealth, complemented by a variety of rock art locations, which substantially enhances our comprehension of the area's history and its interconnected association with paleoclimatic conditions. Through the examination of archaeological sites and artifacts, scholars can determine the temporal and spatial extent of human habitation, providing insight into historical weather patterns and their impact on human societies. Furthermore, the study of rock art offers further understanding of the complex interplay between the civilization of ancient Egypt and fluctuations in paleoclimate, as noted by [1,40].

The rock art discovered in the Kharga Oasis is a significant source of information regarding the paleoclimatic variations and the corresponding human reactions during the wet phase of the Holocene era. Additionally, it can provide insights into the subsequent shift to hyper-arid conditions in recent times. The present investigation is based on the application of prehistoric petroglyphs and rock art discovered in the Kharga Oasis as a significant resource of paleoclimate-derived information. The Kharga Oasis is recognized for its abundant collection of inscriptions, writings, and rock art drawings. A total of 136 archaeological sites were identified, including—for examples- Jabel al-Tair, al-Gaga, Ain al-Labkha, Umm al-Dabadeb, and Ain Amur. In these sites, the artistic portrayals included both animal and anthropomorphic representations. Based on rigorous field research conducted in these archaeological regions, it has been noted that rock art is primarily concentrated on the outskirts of the Eocene plateau, on remote hillsides, and inside caverns.

The predominant method of depicting landscapes in natural settings involves the utilization of the pecking technique on sandstone formations. Nonetheless, this form of rock art necessitates additional scrutiny through the lenses of historical, geographical, and archaeological analyses. However, the present research provides evidence for the progressive transition of climate conditions from moistness to dryness, as suggested by the portrayals of the terrain.

## 3. Results and Discussion

The main objective of the current study was to evaluate the susceptibility of the ancient Egyptian civilization to alterations in climate, examine the reactions of the civilization to these modifications, and investigate plausible causal connections between fluctuations in paleoclimatic conditions and environmental incidents. This investigation adopts theoretical frameworks that revolve around the geographic setting of the Kharga Oasis, with a particular emphasis on scrutinizing the impacts of paleoclimatic variations.

In recent years, there have been several scholarly investigations that have concentrated on the restoration of paleoclimatic variations in the Western Desert of Egypt [1,1,7,25,34,38,39,41,42,44–46]. The aforementioned researches have employed various metrics such as radiocarbon dating of human habitation, geomorphological scrutiny encompassing the detection of lacustrine deposits, and the scrutiny of plant macro remains to reconstruct the crucial ecological circumstances that support human sustenance. In this study, rock art evidence was employed to examine the effects of paleoclimate fluctuations on the Kharga Oasis [7,11,47,48].

### 3.1. Paleoclimate Characteristics of the Kharga Oasis

The Kharga Oasis was predominantly subject to arid climatic conditions during the Holocene prehistoric period. The climatic conditions in the Kharga Oasis from 4000 to 9000 BC are illustrated in Figure 2, which displays a cyclic pattern of dry and wet episodes. The prevailing climatic condition during the aforementioned period was predominantly wet, as observed between 6000 and 7200 BC. Preceding and after this pluvial episode, the Kharga Oasis encountered reduced precipitation, and occasionally, arid circumstances [49].

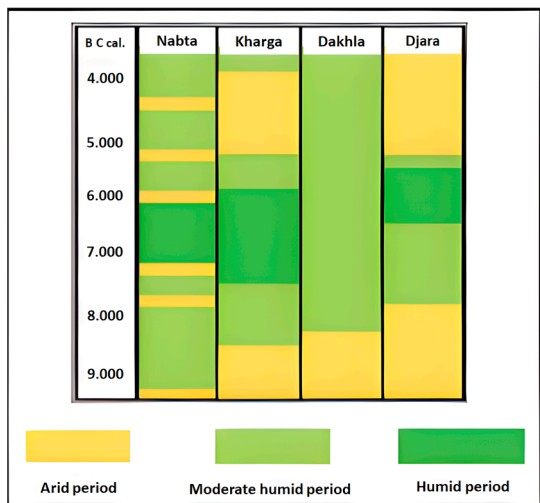

**Figure 2.** Reconstructions of paleoclimate over four zones across the Egyptian Western Desert. Source: [49].

The climatic patterns that have been observed in the Kharga Oasis are consistent with analogous trends that have been documented in other areas of the Western Desert (Figure 2). As illustrated, the interval spanning from 8000 to 9200 BP characterized by heightened humidity and precipitation, represents a noteworthy epoch in the history of the Kharga Oasis. These humid climatic conditions may provide a conducive environment for the proliferation and maturation of flora, consequently fostering a wide range of ecological populations. The probable influence of water resources on human settlements, agriculture, and societal dynamics in the area during this era is noteworthy. However, it should be stressed that wet periods were not incessant, but rather interspersed with arid intervals preceding and succeeding them. Approximately 7300 BP, there was a significant increase in arid conditions, leading to the migration of populations from regions lacking access to groundwater resources, this has been documented by [1,2,11,39,40,50].

During the period ranging from 8000 to 9000 BP, human settlements had become firmly established across the expanse of the Western Desert. During the Middle Holocene epoch, specifically between 7300 and 8000 BP, a phase of pastoralism was observed, which was evidently influenced by the Near East. This moist phase was commonly ascribed to amplified solar activity induced by orbital forcing, leading to elevated atmospheric temperatures, escalated precipitation, and denser vegetation, thereby facilitating human habitation in the Kharga Oasis (refer to Figure 3C).

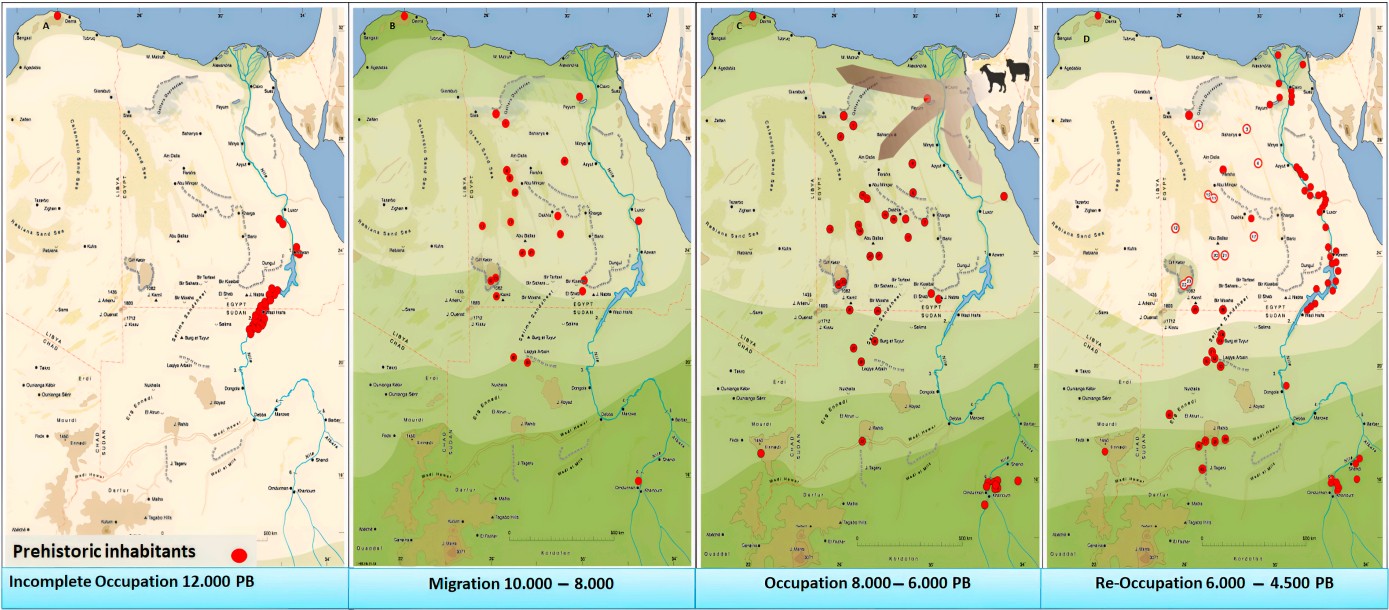

**Figure 3.** Distribution of the Egyptian settlements in the Western Desert throughout the Holocene. Source: [1,11,35,51].

During the Middle Holocene epoch spanning from 5500 to 7300 BP, there was a notable trend towards regionalization, as individuals and communities relocated to regions with a consistent water supply and prioritized agricultural activities in the Nile Valley. Approximately 7300 BP, the onset of drought and desertification in the Kharga Oasis was triggered by a decrease in monsoon rainfall. In ancient times, human populations were compelled to relocate to areas where surface water and precipitation were still conducive to survival, as depicted in Figure 3D. This resulted in the settlement of the Nile Valley. The emergence of Pharaonic civilization in the Nile Valley approximately 5500 BP was concurrent with the return of arid conditions throughout Egypt, as noted by [1,5,6,10,45,46].

The paleoclimate fluctuations in the Western Desert have been documented in earlier studies. According to Neumann [52], there was a variation in precipitation levels during the Holocene period, ranging from 50 mm/year at the onset of the era to 100 mm/year during the Middle Holocene. Hoelzmann [53] demonstrates that the zenith of lake sedimentation took place circa 6000 BC, suggesting a climatic apex of the moist phase of the Holocene. The climatic conditions during the Early/Middle Holocene were characterized by fluctuations between arid and semi-arid phases, indicating a certain degree of instability. According to Pachur and Hoelzmann [53], there was a gradual reduction in precipitation from the western to eastern regions and from the northern to southern regions, ultimately resulting in the current state of hyper-aridity.

### 3.2. Evidence on Paleoclimatic Variability as Derived from Prehistoric Rock Art

The Kharga Oasis provided a conducive habitat for prehistoric communities, as well as for the growth and sustenance of plant and animal life. The early Holocene epoch, spanning from 8500 to 7000 BC, is believed to have had a divergent impact on the river

Nile valley and the Western Desert of Egypt [15,52]. The former region is thought to have become unsuitable for human habitation during this period, while the latter underwent a significant shift from a hyper-arid desert to a savanna ecosystem. This transformation was marked by the emergence of various forms of flora and fauna, human settlements, as well as the formation of lakes and rivers. On the contrary, according to Kuper and Kröpelin's [1] findings, the ancient Nile River was not conducive to human habitation.

The temporal arrangement of rock art depictions plays a crucial role in comprehending the dispersion of ancient communities and the characteristics of their ways of life. The rock art sites situated in the Kharga Oasis are predominantly situated in the central and northern regions, with a notable aggregation in the western area of Ain El-Labkha and along the Darb Ain Amur, which is the route linking the Kharga to the Dakhla Oasis [40]. The locations cover diverse eras of ancient Egyptian prehistory and comprise a broad spectrum of artistic representations, such as portrayals of topography, footwear, watercraft, imprints, everyday activities, fauna, written characters (including Demotic, Hieratic, hieroglyphic, and subsequent scripts), and mathematical designs. In accordance with these findings of Zboray [44], our observations demonstrate that a significant proportion of rock art depictions in the Kharga Oasis pertain to livestock and pastoralism, and a few of them portray the pastoral horizon.

The current investigation acknowledges rock art and petroglyphs as the foremost and dependable means of observing the phases of paleoclimatic transformation. The Kharga Oasis boasts a plethora of sites that feature inscriptions, writings, and rock art drawings. Among these sites, Jabel al-Tair, Al-Gaga, Ain Al-Labkha, Umm El-Dabadeb, and Ain Amur are considered particularly significant. Empirical investigations conducted on these archaeological sites have revealed that petroglyphs are predominantly located on the periphery of the Eocene plateau, in remote hillside locations, and within subterranean caverns.

The rock art discovered in the Kharga Oasis offers a comprehensive depiction of various fauna, such as mammals, reptiles, birds, fish, amphibians, and insects. The creatures in question held a noteworthy position in the spiritual convictions and everyday existence of the local populace, very similar to the portrayal of the universe in subsequent Egyptian temples and tombs. The ancient inhabitants' daily lives and spiritual beliefs revolved around animals, as shown in these depictions. The inclusion of their depiction in rock art suggests a correlation with significant alterations in the climate, as per the findings of Ikram [5,40]. The depiction of men leading giraffes on leashes and giraffes in a recumbent position is presented in Figure 4. The artwork portrays two elephants, albeit on a smaller scale than the giraffes, suggesting their artistic depiction.

The existence of culinary implements (as depicted in Figure 5E) serves as an indication of urbanization in the Kharga Oasis, signifying the emergence of established societies. The unearthing of a subterranean temple beneath sand dunes (depicted in Figure 5F) implies noteworthy variations in climate, encompassing alterations in wind patterns and velocity that could have played a role in the interment and conservation of the temple. The data presented in Figure 5G,H suggest that the density of human bones and skulls can serve as an indicator of densely populated residential communities. This, in turn, may suggest the presence of favorable conditions such as the availability of water sources and a climate that is conducive to human habitation. Additionally, the existence of rainwater gathering sites on the Eocene plateau and vestiges of dried-up valleys (as depicted in Figure 5I,J) provide substantiation for the occurrence of intensified ancient precipitation in the area.

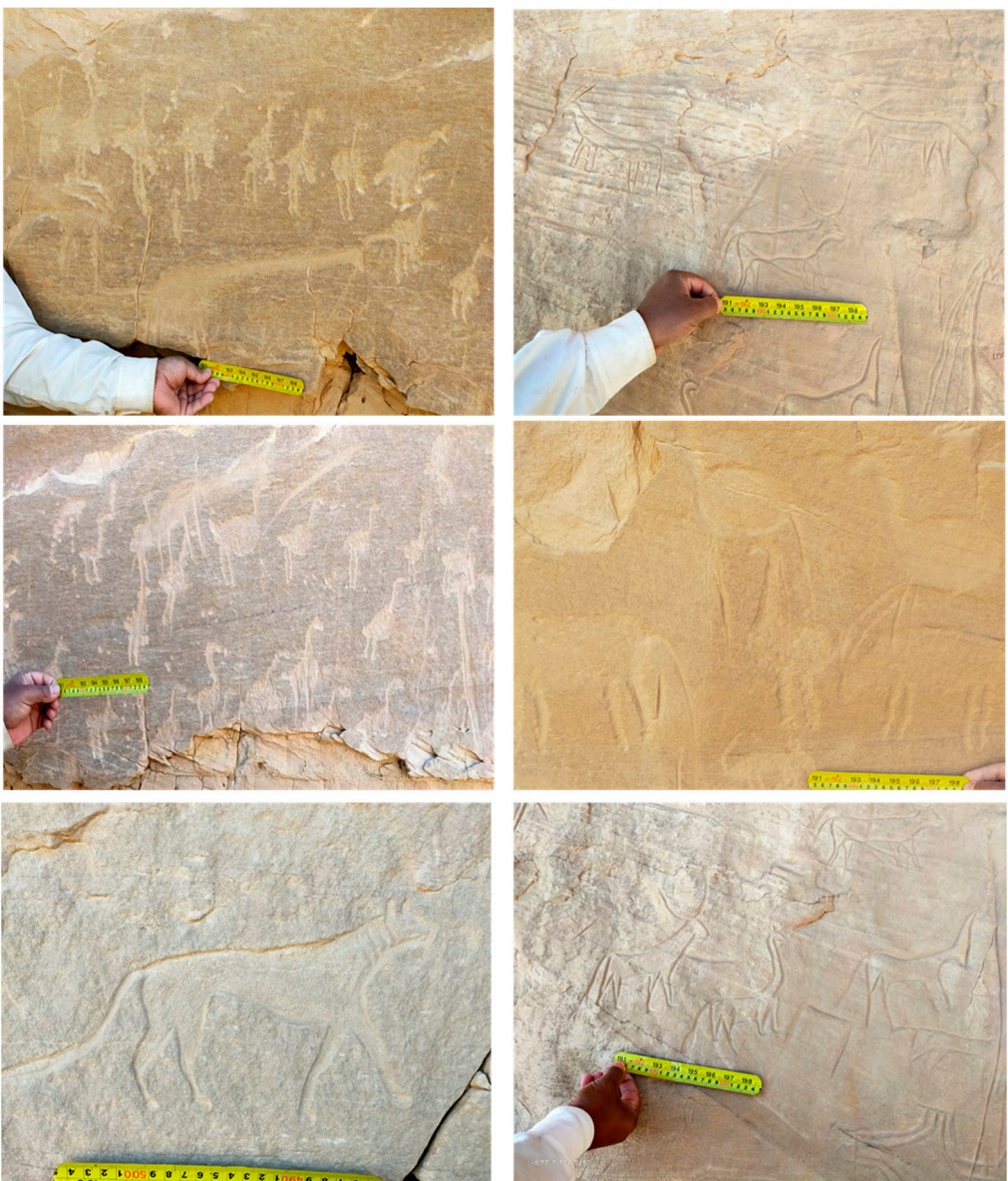

**Figure 4.** The gallery range of animal species represented in the Kharga Oasis, with the record of fauna found in the rock art of the Kharga Oasis showing a common grouping of lioness, Oryx, and giraffe.

In the Kharga Oasis, the observation has been made that dense clusters of domesticated animals flourished in climates that were more humid than the current arid conditions. Currently, the arable land in the area has been spared from the adverse effects of both drought and desertification. The data interconnectivity indicates the existence of a savannah-type ecosystem characterized by enduring water reservoirs and periodic vegetation comprising of trees and grasses. According to [5,44], the climatic conditions were conducive to the sustenance of grazing livestock as well as the survival of wild animals that relied on water sources. Giraffes and elephants indicate a savannah-like environment with water and grazing vegetation. This suggests that climate change affected animal populations and the human communities that relied on them. In general, the illustrated animal life portrays a customary Holocene ecological system, which mirrors the ancient environment of the Kharga Oasis.

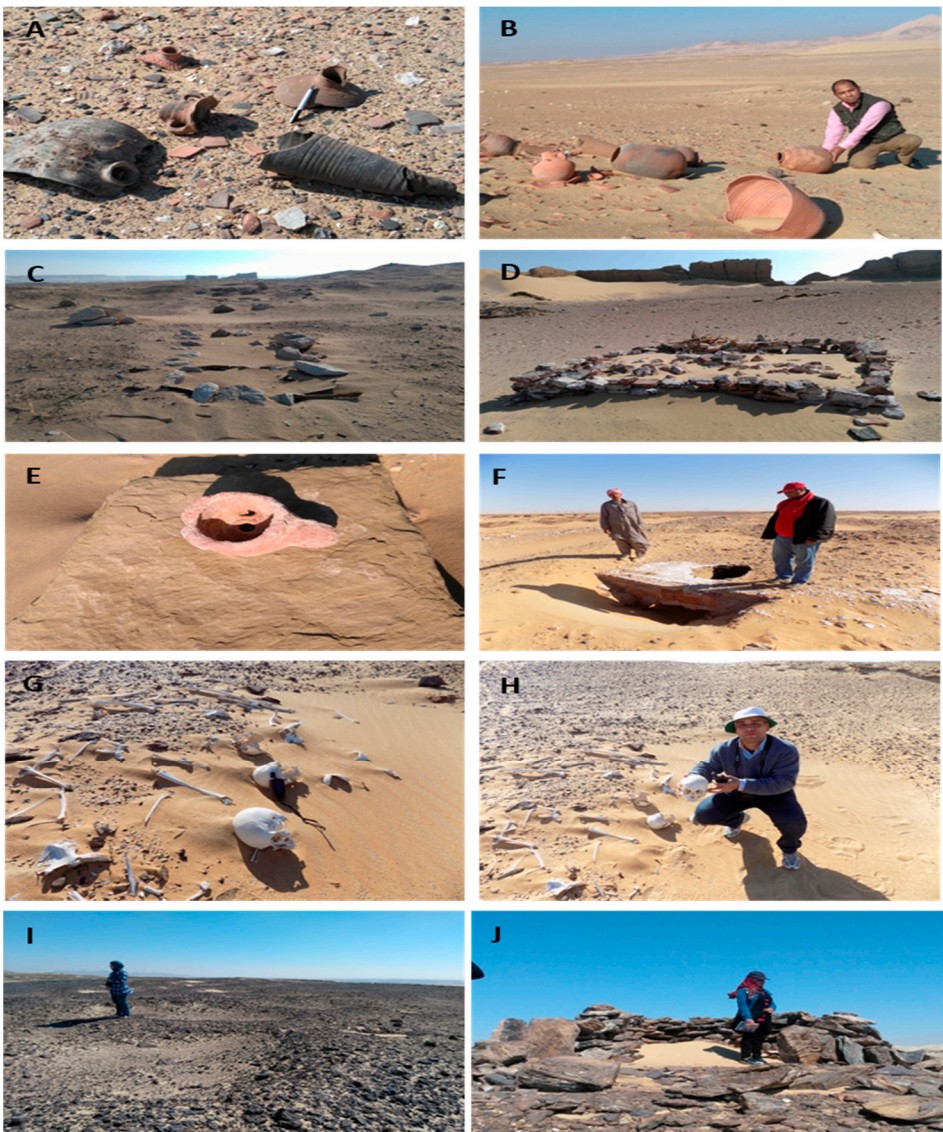

**Figure 5.** A collection of cultural remains that have been carefully chosen from the Kharga Oasis. The presented compilation exhibits a diverse array of artifacts and archaeological remnants, which function as tangible evidence of historical climate fluctuations within the given geographical area; (**A**,**B**). Remains of pottery vessels, (**C**,**D**). irrigation wells, (**E**,**F**). cooking tools, (**G**,**H**). Skeletal remains, and (**I**,**J**). security mentoring points.

Additionally, the rock art paintings serve as empirical proof of the prehistoric population movements that occurred within the Kharga Oasis. The visuals illustrate the mobility of limited clusters of individuals in reaction to changes in both temperature and precipitation. The artwork's inclusion of certain animal species suggests more humid climates than the present arid climate. The rock art implies irregular seasonal migrations between the Kharga Oasis and the Nile Valley, driven by extreme climate changes. The migratory trajectory presents a notable divergence from the spatial and archaeological data pertaining to the historical civilization of Egypt. Figures 6 and 7 depict various images, including public triangles, a boat, Oryx, goats, longhorn cattle, a bird, and a man resting on his knee. While early rock art portrays wild cattle, it is probable that most cattle depictions originate from the era following their domestication, specifically during the Neolithic period circa 8000 BP [1,5,39].

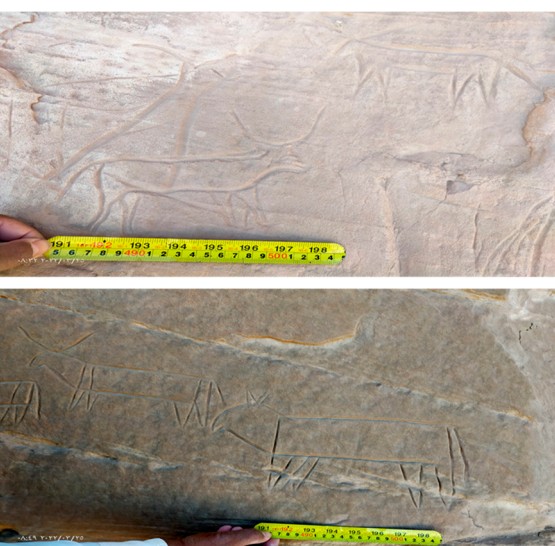

**Figure 6.** Petroglyph from the northeastern sites of the Kharga Oasis showing different types of chattels.

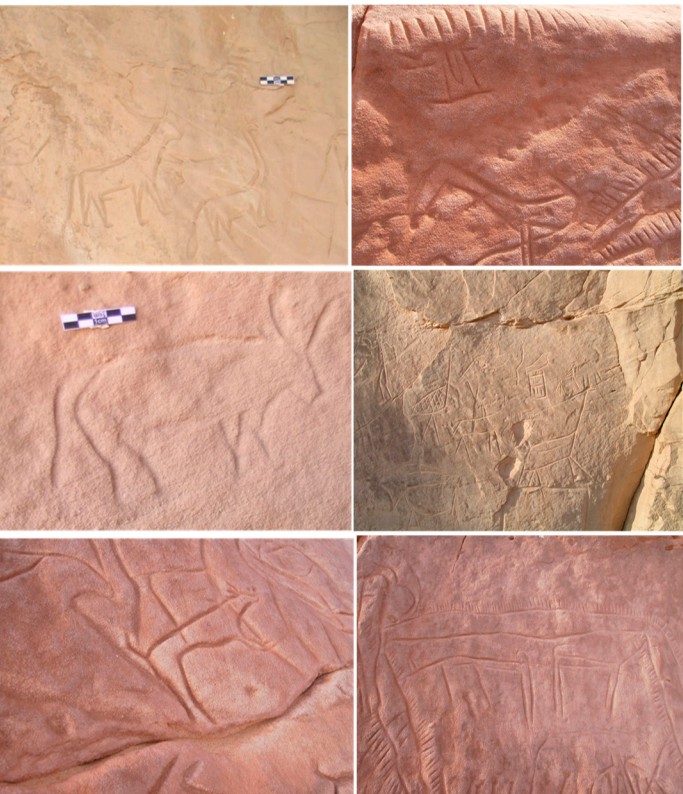

**Figure 7.** Petroglyph from the central sites of the Kharga Oasis illustrating different types of chattels.

It is noteworthy to mention that animal depictions from epochs subsequent to the Holocene epoch comprise of camels, horses, donkeys, and conceivably waterfowls such as ducks and geese. Most of these depictions were executed through the process of engraving. However, instances of the pecking technique and a fusion of methodologies were also detected, leading to the creation of enigmatic configurations (Figure 8).

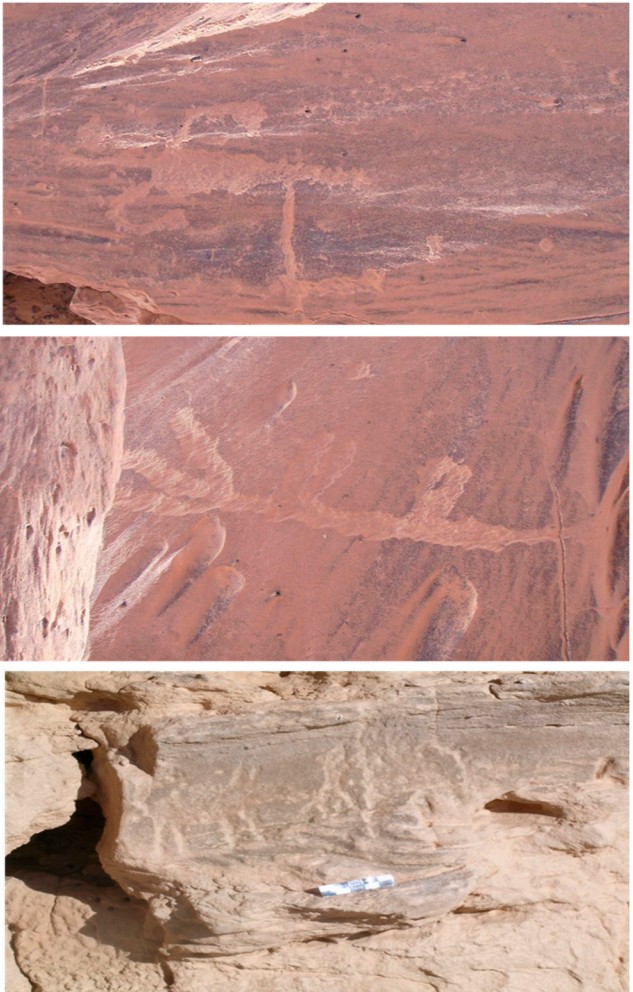

**Figure 8.** An illustrative example of the depictions of the pecking technique, a traditional technique used in the creation of petroglyphs in the study domain.

Based on the extrapolation of the prehistoric landscape representations, it is apparent that the Kharga Oasis held great importance as a site that was abundant in depictions of the surrounding landscape. The markings and illustrations appear to suggest the implementation of a methodology that involves the engraving of geometric figures (such as equilateral triangles, quadrilaterals, and numerous beams) that hold symbolic connotations related to agriculture, reaping, and a comprehension of daily cycles (refer to Figure 9). The prehistoric era is credited with the creation of three-dimensional expressive representations of road paths, maps, and guide signs. Furthermore, a number of inscriptions that showcase footprints and sandals are also observable (refer to Figure 10).

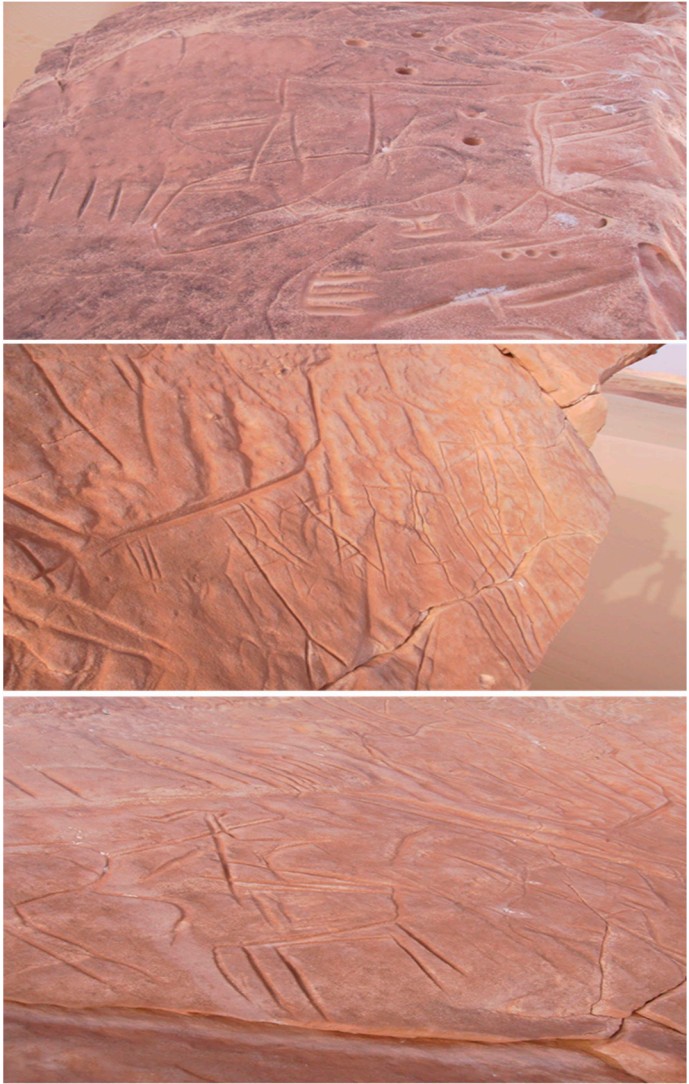

**Figure 9.** Expressive three-dimensional representations of the routes, maps, and guide signs.

The paleoclimatic conditions of the area were impacted by monsoonal activity, which was distinguished by substantial precipitation during the summer season. The movement of the monsoon caused savannah-like environments that turned the Kharga desert into a habitable region, and prehistoric humans soon settled there. The Nubian sandstone formation caused the emergence of fresh and shallow lakes, temporary small valleys, and well-tolerated vegetation in the Oasis. Currently, the area is covered by extensive sand dunes. This information is supported by Nicolland Kuper and Kröpelin [1]. Subsequent studies have revealed that during earlier periods, the intensity of monsoon front rainfall was higher compared to the present. Additionally, the sand dunes located within the Kharga Oasis were primarily covered with vegetation. The available data indicate that the climate in the southern Kharga Oasis was characterized by an ITCZ rain front during the period spanning from 6400 to 9500 BP. According to Linstädter and Kröpelin [54], rainfall was infrequent but intense, with an estimated occurrence rate of approximately four times per century. During the early to middle Holocene epoch, the Kharga Oasis was marked by the existence of a multitude of shallow freshwater bodies [8,24,33,35,55–58]. The presence of these freshwater bodies on the surface proved to be a significant draw for both livestock and agricultural laborers. The ascertained fauna inhabiting the vast freshwater lakes encompassed rodents, hares, gazelles, and a limited number of carnivorous species. The depiction of various animal species such as ostriches, deer, elephants, giraffes, lions, and leopards has been uncovered through rock art paintings. The gradual decline in the

efficacy of the monsoon's copious precipitation led to its complete cessation circa 5000 BC. This, in turn, caused the Kharga Oasis to be overrun by sand dunes and the vegetation to perish, as reported by [1,39,40,44].

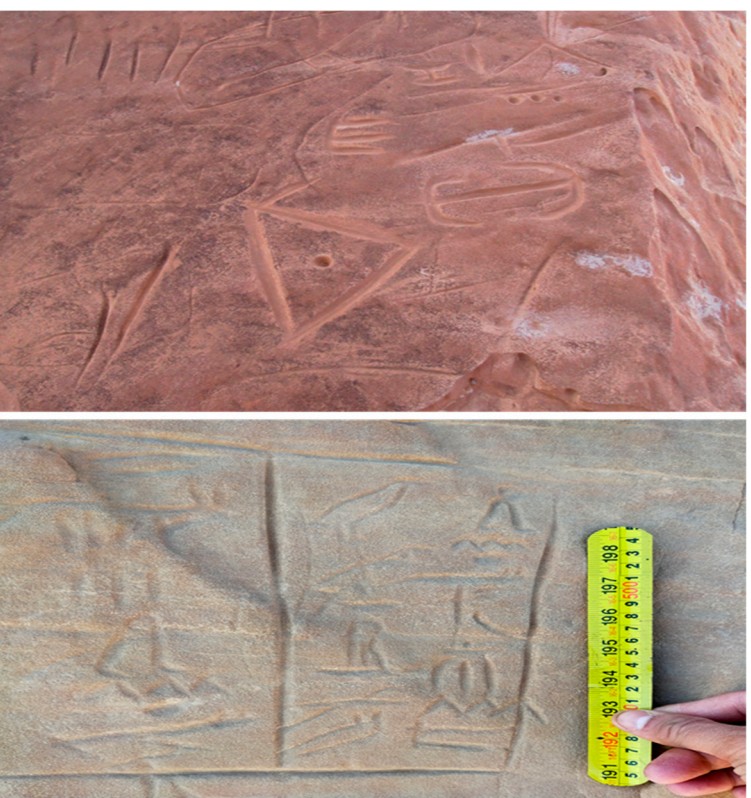

**Figure 10.** An array of distinct styles employed in the depiction of landscapes across various periods and cultural contexts in the Kharga Oasis.

During the Mid-Holocene epoch, a significant increase in aridity occurred throughout the majority of the Kharga Oasis, resulting in the displacement or extinction of various flora and fauna, which were forced to relocate to particular regions. The gradual migration of shepherds from the Sahara through the Nile Valley commenced circa 4500 BC, coinciding with the advent of the dry season. This phenomenon is estimated to have taken approximately 1000 years to complete [2,24,47,55]. As per the findings of [8,59], the Late Holocene epoch witnessed noteworthy alterations in the paleoclimate, which played a crucial role in the exceptional development of grazing practices in the Kharga Oasis. These practices have been documented in prehistoric records. According to McDonald [50], the genesis of ancient Egyptian pastoralism can be traced back to the desert ecosystem.

To summarize, it is noted that the literature review emphasizes the importance of prehistoric animal representations in rock art as evidence of past climate change in Egypt. These images reveal ancient fauna, climate, and human responses to environmental changes. As such, our results can provide insights into the varied fauna that inhabited the region, their cultural relevance, the impact of climatic variations on their depiction, and the migratory patterns of ancient communities in response to environmental changes. The representations offer significant perspectives regarding the historical environment and the dynamics of human-animal relationships in the area. Additionally, although landscape rock art is not a naturally occurring phenomenon, it is acknowledged as a means by which the activities of ancient Egyptians were expressed, possessing a sense of efficacy. In the given context, the landscape is perceived to possess a degree of stability amidst societies that are constantly evolving and actively contributing to the emergence of new interpretations, as per Tilley's perspective [60].

However, herein, it is noteworthy to indicate that while ancient inscriptions provide valuable insights into the environmental conditions of previous epochs, establishing the chronological framework of petroglyph production remains a challenge [8,35,61].

One common approach is to analyze superimposition and stratigraphy. In instances where petroglyphs exhibit overlapping or superimposed elements, it is commonly deduced that the underlying artistic depictions predate the ones on top, thereby providing a sequential chronology. An alternative approach, known as associative dating, involves the use of petroglyphs' spatial relationship to other archaeological materials or contexts, such as artifacts or organic remains, which can be dated utilizing techniques such as radiocarbon dating [47].

In the same context, stylistic analysis provides an additional approach to determine the age or time period of a given artifact or text [36,47]. Through the process of comparing the stylistic elements and recurring themes found in petroglyphs with other artworks or inscriptions that have been accurately dated, researchers have the potential to establish rough estimates regarding the time periods in which these petroglyphs were created. In our work we draw comparisons between the images and artifacts originating from the Nile Valley, which possess more precise chronological attributions. Occasionally, the portrayal of particular animal species can offer valuable indications for dating purposes. In the course of our investigation encompassing 136 rock art sites located within the Kharga Oasis, we systematically classified the rock art manifestations into four discernible temporal periods. During the hunting stage, which occurred approximately 12,000 years ago, there is evidence of artistic representations depicting living creatures with disproportionate sizes, including elephants, hippos, rhinos, and various abstract designs. The shepherd stage, which emerged approximately 9000 years ago, placed significant emphasis on the practice of animal domestication. During the period approximately 8000 years ago, there was a notable prevalence of horse-drawn wooden carts. The proliferation phase of camels occurred approximately 7300 years ago subsequent to the southward displacement of the Intertropical Convergence Zone (ITCZ). Following this procedure, we found that our findings are consistent with the research conducted by other scholars such as [13,16,16–18,20,48,62].

### 3.3. Drivers of Paleoclimatic Variability

Over the course of the last twelve millennia, the distinct features of the Kharga Oasis have been notably impacted by fluctuations in climate. Extensive interdisciplinary research has examined the mechanisms behind these alterations and the corresponding adaptive measures implemented by the indigenous populace. The displacement of the ITCZ front is a frequently cited phenomenon in the African tropical and sub-equatorial regions, which is mostly linked to variations in Holocene rainfall. This process is attributed to the latitudinal migration of the ITCZ front. This displacement is extensively documented in global archives such as lacustrine records, ice cores, and marine sediments.

Numerous studies attributed the alterations in the ITCZ, and thereby paleoclimate over the Sahara, to orbital forcing [12–14,48]. The millennial orbital forcing played a key role in the gradual southward displacement of the ITCZ position in the Northern Hemisphere's summer (Figure 11a,b). This resulted in weakened monsoon systems and increased frequency of drought and desertification in the study area. As a result, the ITCZ boundary and associated monsoonal mechanism underwent a southward displacement of approximately 800 km, as illustrated in Figure 12. Many studies suggested a high inclination of the Earth's orbit approximately 10,000 BP, succeeded by a reduction of approximately one degree of latitude. The ITCZ location is linked to the Earth's zone of maximum insolation, which is influenced by the Earth's orbital inclination, as noted by Gasse et al. [36]. This orbital forcing affects the insolation of both hemispheres, resulting in the latitudinal shift of the ITCZ, which moved approximately one degree north and south during the early Holocene. According to [35,57,63], the ITCZ advanced further northwards at the beginning of the Holocene, leading to rainfall decrease (50–80%) northwards from evergreen rainforests towards the Western desert. During the early Holocene, the ITCZ

moved approximately 800 km northward, reaching 20–21° N. During the Holocene's onset, the authors of [29,42,59,62,64] observed that the westward winds propelled the incursions of moist air masses from Mediterranean latitudes, which reached a minimum of 24° N and 30° E.

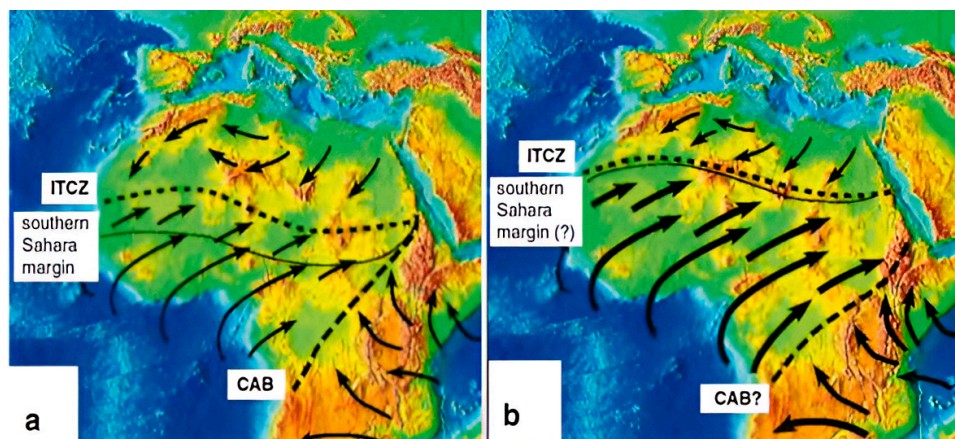

**Figure 11.** (**a**) Current positions of the ITCZ movement in summer with the southern margin of the Western Desert and wind directions, and (**b**) the derived early ITCZ movement and winds in the winter of Holocene. Source: [59].

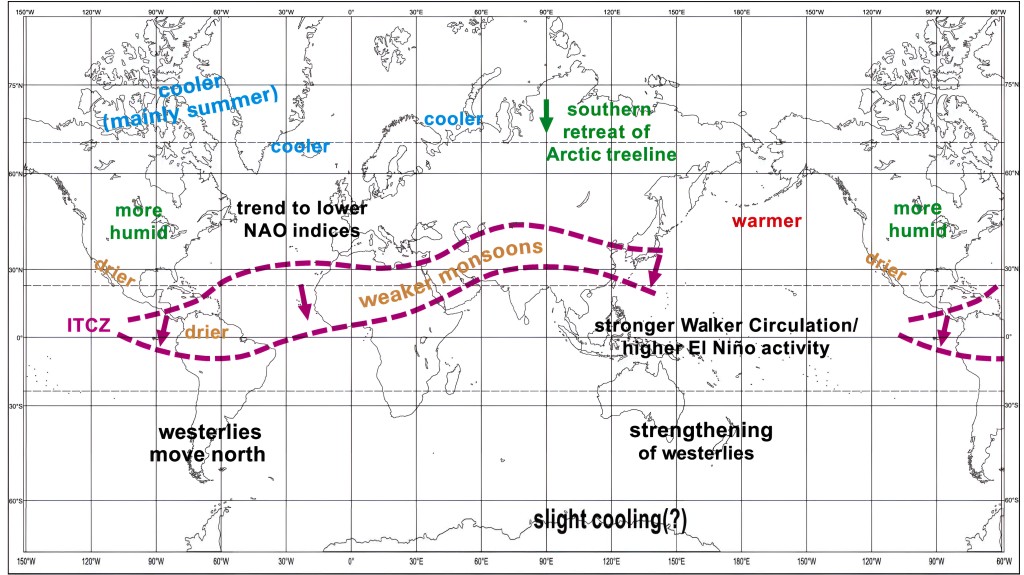

**Figure 12.** Spatial combination of the world climate change during preindustrial times (AD w1700) in comparison to the MH (w6000 cal years BP). Source: (in [35]—Figure 18. Spatial synthesis: global climate change for the preindustrial period (AD ~ 1700) compared to the MH (~6000 cal years BP).).

Subtropical to tropical climate systems are dominated by Hadley cells, with hot air rising at the ITCZ front and cooler air descending approximately 30° North and South, flowing back toward the ITCZ as the Easterly Trade Winds. The latitudinal oscillations of the ITCZ lead to periodic shifts towards the north and south, thereby causing the arid regions to receive precipitation during summers and the Mediterranean Coast during winters. The northward migration of the ITCZ in the summer season controls the annual precipitation levels. Specifically, when the ITCZ circulation extends further north, there is a longer and more intense rainy season [56,57,59,65].

For the Kharga Oasis, some studies revealed pronounced paleoclimate changes. The paleoclimate phenomenon observed in the Kharga Oasis was impacted primarily by two

discrete systems, namely, the winter rains originating from the north and west, and the summer rains originating from the African tropical monsoon system. After 12,000 BP, summer rains from the south allowed the Egyptian Western Desert to be settled beyond the Nile River, with attempts to establish settlements along the river impeded by the risk of flooding (Figure 3A). The hyper-arid region underwent a climatic transformation, leading to the emergence of a savannah-like environment that was promptly inhabited by prehistoric settlements. Between 8500 and 7000 BC, during the Ancient Holocene, there was reoccupation marked by hunter-gatherers, pottery, and domestic cattle (Figure 3B). The aridification observed in the Kharga Oasis during the Mid-Holocene, as indicated by the records of lake and vegetation, is believed to be caused by the displacement of the ITCZ towards the south. This theory has been proposed by [1,2,59,65,66].

Changes in the paleoclimate of the Kharga Oasis have had a significant impact on human communities throughout history. This variability in climatic conditions would have presented difficulties for human communities, impacting the accessibility of resources, methods of subsistence, and the general capacity of societies to adjust to shifting environmental conditions. However, the similar climatic pattern observed in other zones of the Western Desert may suggest a regional-scale influence (e.g., ITCZ shift) on the climatic dynamics during this time period. The alterations in climate patterns within the Kharga Oasis have had noteworthy consequences on human endeavors in the surrounding areas of the Nile.

The historical trajectory of the Kharga Oasis has been significantly influenced by the climatic oscillations that have occurred over the course of several millennia. Approximately 1100 years ago, it is believed that a prolonged period of drought lasting two centuries may have played a role in the demise of civilizations inhabiting the oases. The Holocene period witnessed a prolonged and variable rainfall intensity, coupled with the southward shift of the ITCZ, which has had a significant impact on the migration and disappearance of civilizations within the oases. This phenomenon has been extensively studied by [11,36,51,57,67]. Prior research has indicated that fluctuations in climate were responsible for the dispersion of settlements throughout the expansive Western Desert. This phenomenon was brought about by heightened levels of monsoon precipitation and the consequent accessibility of water resources, which in turn facilitated agricultural practices and fostered stability within the Kharga Oasis. As a result, the region became endowed with an abundance of organic matter that was essential for successful farming. These settlements also served as a response to the hazards associated with Nile River flooding and swampy conditions in the southern part of Egypt.

Based on a visual examination of rock art discoveries and corresponding data indicating the shift of the ITCZ, it can be deduced that a noteworthy migration took place in the Kharga Oasis during the final transition from the last glacial period to the Holocene epoch, in response to the ITCZ's swift southward movement. The rock art's depictions of livestock and the heightened density of vegetation indicate an average displacement of the ITCZ of over 800 km. This suggests a swift escalation in monsoon rainfall. The abrupt intensification of the African monsoon during the early Holocene period was triggered by the swift escalation of temperatures and the reduction in ice mass in the high latitudes of the Northern Hemisphere [36].

The observed decrease is construed as a progressive movement towards the south of the ITCZ in the course of the summer season in the Northern Hemisphere, which is instigated by a reduction in the amount of solar radiation received during the summer. The absence of a discernible reduction in monsoon precipitation during the Mid-Holocene, which is a phenomenon frequently detected in other monsoon records from Africa and India, was noted. Bradley and Diaz [68] found that the high-frequency fluctuations in monsoon precipitation coincided with the global Holocene patterns.

### 3.4. Impacts of Paleoclimate Changes on Ancient Civilization

The analysis of past climate variations and their impacts on ancient civilizations presents a captivating viewpoint through which to study past epochs. The Kharga Oasis, one of the primary oases situated in the Western Desert of Egypt, offers a significant opportunity for investigating the potential influence of climatic fluctuations on human societies. The issue of water resource availability has become a significant concern within the framework of paleoclimate change. The utilization of freshwater resources by the residents of the Kharga Oasis, very similar to other inhabitants of oases, played a crucial role in supporting a wide array of activities, including agriculture, potable consumption, and routine tasks. Therefore, the potential ramifications of a shift towards a drier climate or variations in groundwater levels encompass a reduction in agricultural productivity, restricted access to water resources, and the potential for societal turmoil. Empirical evidence suggests that the fluctuation of climatic patterns in ancient civilizations, particularly instances of decreased precipitation, had a noticeable influence on the development and complexity of irrigation systems [69,70]. The civilizations in the region were motivated to adapt their practices in order to sustain a consistent level of agricultural productivity in light of these transformations. It is pertinent to acknowledge that the ancient Egyptian society in oasis regions, such as the Kharga, relied heavily on agricultural practices as a fundamental aspect of their societal structure. Paleoclimatic changes have indeed had an impact on the viability and productivity of crops, as they have been associated with alterations in precipitation patterns and shifts in growing seasons. Various factors, including prolonged periods of drought, played a significant role in the occurrence of famines and the subsequent scarcity of food. Moreover, in the event of a decrease in agricultural productivity, there exists the possibility of an escalation in malnutrition rates, consequently making the population more susceptible to a variety of diseases. One notable illustration is the decline and collapse of the Akkadian Empire in Mesopotamia, widely acknowledged as one of the earliest instances of a genuine empire in human history. This empire witnessed its demise approximately 4200 years ago. A multitude of scholarly investigations have demonstrated a significant association between the downfall of this particular civilization and abrupt alterations in climatic circumstances. The sedimentary records obtained from the region provide evidence of a pronounced and abrupt drought event that occurred during the same time period as the decline of the empire [71]. The protracted duration of the drought, lasting for several decades, is anticipated to have had significant ramifications on the agricultural industry, leading to societal instability and eventual collapse. Furthermore, it is worth noting that during the Late Bronze Age in the Eastern Mediterranean region, there were complex and multifaceted interactions that took place among the Mycenaean kingdoms, the Hittite Empire, and the New Kingdom of Egypt. In approximately 1200 BCE, there was a notable occurrence of a substantial societal collapse that had far-reaching impacts on various regions. The investigation of paleoclimate data, involving the examination of tree rings, indicates the presence of a significant drought within the designated temporal period. The dissolution of these formidable states is widely believed to have been influenced significantly by climatic factors, in addition to other socio-political and military factors.

The Kharga Oasis experienced socio-political disruptions as a result of climate change-induced stress, particularly in relation to essential resources such as food and water. The limited availability of resources possesses the capacity to function as a catalyst for conflicts, consequently destabilizing established societal structures and potentially leading to migrations or incursions by groups enticed by the plentiful resources of the Oasis. Moreover, the Kharga Oasis played a pivotal role in trade networks during specific periods of ancient Egypt due to its strategic geographical location. Climate changes could have had an impact on trade routes, potentially resulting in their alteration or reduced viability due to the increased challenges associated with traversing arid or unpredictable terrains [31]. Furthermore, the occurrence of climate change possesses the capacity to lead to a decline in the production of certain commodities that are frequently exchanged, such as various types

of grains. This has the potential to not only change the sources and destinations of trade, but also potentially modify the prevailing trade routes that are utilized [31].

From a cultural perspective, the ancient Egyptians demonstrated a profound spiritual bond with the Nile and their environment, causing them to interpret adverse climatic fluctuations, such as extended periods of drought, as possible indications of divine discontent. This perception possesses the capacity to initiate modifications in religious practices, rituals, or even the pantheon of revered deities. Throughout different epochs in history, the intricate relationship between individuals and their environment, specifically noteworthy factors such as the climate, has been intricately linked with their spiritual beliefs and ceremonial practices. In various historical societies that were predominantly agrarian, the fluctuations in climatic conditions, such as prolonged droughts or unanticipated patterns of precipitation, exerted a substantial influence on the productivity of crops. These modifications have the potential to lead to cultures giving more importance to or raising the significance of deities associated with rainfall, fertility, or agricultural activities. To ensure a bountiful harvest, it is reasonable to consider implementing or strengthening rituals that are designed to appease these divine beings. An exemplification can be observed in the ancient Egyptians, who held a deep reverence for the yearly inundation of the Nile due to its significant impact on their agricultural methods. Consequently, the deity Hapi, who held a strong association with the Nile River and its yearly flooding, was revered, and religious rituals emerged in accordance with the natural patterns of the river. Furthermore, in specific circumstances, societies have been known to create new divine beings as a means to represent and address recurring or increasingly severe climatic adversities. For instance, in a society characterized by persistent drought conditions, it may be deemed appropriate to institute a deity that is intricately associated with the occurrence of precipitation or the management of water reservoirs. Furthermore, the importance of sacred sites may undergo alterations due to climate modifications. The depletion of water in a previously esteemed water source may lead to a decline in its religious importance, while the appearance of a new water source in a distinct area has the potential to become the central focus of worship [8,72–74]. As a result of this shift in religious emphasis, it is possible that new temples and pilgrimage routes may be established. In general, it is anticipated that the communities residing in the Kharga Oasis will develop new cosmological narratives in order to explain significant shifts in climate. Consequently, religious beliefs and the corresponding practices may undergo substantial transformations in response to these paleoclimatic changes.

### 3.5. Study Limitations and Future Outlook

Petroglyphs serve as invaluable portals into the historical records of ancient human civilizations, providing insights into their belief systems, cultural practices, and daily lives. The indispensability of digitizing and analytically studying these relics arises from their susceptibility to both environmental decay and human interference. Traditionally, researchers have primarily relied on observational techniques and direct assessments, particularly in geographically significant areas with a rich historical background, such as Egypt. Nevertheless, the preservation of Egypt's esteemed cultural heritage has encountered numerous obstacles, ranging from insufficient financial backing for conservation initiatives to the vast expanse of the Western Desert, which encompasses a vast array of rock art spanning an area of 680,000 square kilometers. Furthermore, the proliferation of bureaucratic complexities, exacerbated by the socio-political consequences following the January 2011 revolution, have resulted in the discontinuation of various conservation initiatives, particularly those focused on the Kharga Oasis.

The study of Egyptian cultural heritage has traditionally relied heavily on direct observations as a primary methodology. The aforementioned practice facilitated a particular form of close connection with the objects, yet it frequently resulted in a level of subjectivity, as interpretations could potentially diverge depending on individual viewpoints and biases. Emerging technologies, such as photography, provided a means to visually capture the essence of petroglyphs. However, the limitations of two-dimensional photographs often

hindered their ability to fully capture the depth and intricacies of these engravings. It is worth noting that certain variables, such as inadequate lighting conditions, have the potential to distort the quality of these images. Additionally, employing direct techniques such as rubbings may have the drawback of being intrusive [45,75]. In light of the aforementioned challenges, this study proposes the utilization of knowledge derived from previous scholarly endeavors and investigations, with a particular emphasis on the imperative to protect tangible manifestations of our historical heritage, such as inscriptions, scripts, and emblems that adorn rock art. By integrating insights gained from prior expeditions, our objective is to establish a robust trajectory for the preservation of Egypt's rich cultural heritage.

Currently, there is a resurgence of interest in petroglyph studies due to the implementation of innovative methodologies. One notable application of 3D scanning technology is its ability to accurately capture the intricate details of petroglyphs, enabling digital interactions without the need for invasive methods. The confluence of technology and humanities is exemplified by initiatives such as 3D Pitoti and IndianaMas. These projects utilize various imaging techniques, ranging from basic photography to advanced methods such as reflectance transformation imaging (RTI), highlighting the significant contribution of digital tools in preserving, analyzing, and disseminating information about ancient engravings. These advancements reveal previously undisclosed intricacies, providing deep understanding into the artistry, intentions, and potential chronology of ancient petroglyphs, thus expanding the reach of these ancient engravings to broader populations [69,76].

The integration of technological capabilities and visual discernment in the field of petroglyph research offers a valuable perspective known as visual analytics [77]. This methodology integrates information obtained from 3D scanning, specialized photographic techniques such as reflectance transformation imaging (RTI), and potentially spectral imaging in order to reveal subtle details that surpass human perceptual capabilities. Analytical tools facilitate the examination of extensive datasets, allowing for the identification of recurring patterns or themes. On the other hand, comparative analyses can provide insights into cultural exchanges or common knowledge. The fundamental nature of visual analytics resides in its dual methodology, which serves to enhance our understanding of historical symbols while also ensuring their pertinence in present-day discussions.

In conclusion, recent advancements in technology provide not only improved precision but also a comprehensive and respectful methodology for examining our shared historical narrative. As the global academic community continues to explore the intricacies of human history, the integration of conventional and innovative approaches serves as a guiding beacon. Emerging technologies, such as machine learning and data analytics, have the potential to significantly transform the field of petroglyph research by utilizing large datasets to enhance our comprehension of these ancient rock carvings within their wider socio-cultural context.

## 4. Conclusions and Future Works

Rock art sites constitute a significant yet fragile source of data about ancient times. Due to their unique features and worth, they must be preserved as a non-renewable cultural and psychological treasure challenged by the ongoing abuse of the planet's landscapes. The Sahara formation and expansion dominate the water cycle and climate system across North African, the Mediterranean and West Asian regions.

This study provided a new insight into the ancient history of the Kharga Oasis through an analysis of paleoclimatic conditions and rock art records. The statement highlights the significance of rock art as a valuable resource for investigating paleoclimate change and its wider ramifications. The research affirms that the Kharga Oasis has undergone a shift in climate from one condition to another, characterized by intervals of abundant forests, substantial moisture, and domesticated fauna, which have subsequently been replaced by aridity and desertification. Overall, the main conclusions of the current study can be summarized as:

- In accordance with several cultural and civilizational shifts that have been impacted by alterations in the climate, the civilization that previously flourished in the Kharga Oasis was significantly influenced by paleoclimatic variations.
- The analysis of rock art artifacts and the scrutiny of paleoclimate records have yielded significant revelations about historical events. Through the examination of representations depicted in rock art, a noteworthy occurrence of migration that transpired circa 6000 BP is evidenced. In this era, human migration occurred from the Kharga Oasis towards more reliable water sources in the Nile Valley, motivated by the consequences of paleoclimatic variations.
- The decline in paleoclimatic conditions during the 7th millennium BP led to the migration of pastoral communities from the Kharga Oasis to the floodplain of the River Nile and its environs. The phenomenon of migration resulted in the development of distinct regional identities and cultural formations. The interaction between the Kharga Oasis and the Nile Valley enabled significant communication and commerce, as nomadic pastoral societies adjusted to the arid and unpredictable environment.
- The Kharga Oasis' geographical location has played a pivotal role in comprehending its vulnerability to paleoclimatic variations. The process of aridification in the area has been reinforced by the morphological extension of concentrated monsoon precipitation zones, signifying a shift of the ITCZ front towards the south.
- The analysis of rock art illustrations indicates that the aforementioned migration was a result of the intensification of monsoon precipitation. The African monsoon underwent a sudden intensification during the early Holocene epoch, which was instigated by the escalation of temperatures and the decrease in ice mass in the high latitudes of the Northern Hemisphere.
- The prehistoric petroglyphs discovered in the Kharga Oasis have yielded significant findings regarding the region's climate change progression. Specifically, the animal and landscape depictions depicted in the inscriptions and images have provided valuable insights into the environmental and climatic conditions present during the time period in question.
- The observable erosion rates in the historical temples and fortresses of the Kharga Oasis provide evidence of the impact of climatic changes. Changes in environmental variables, including temperature, wind, and precipitation, have been associated with an increase in erosion, underscoring the significant cultural heritage present in this area.
- This study underscores the importance of rock art and petroglyphs as valuable sources of historical information, enabling the investigation of paleoclimatic patterns and their interplay with the physical terrain. Through a meticulous analysis of these artistic manifestations in the Kharga Oasis, a more profound comprehension of the climatic variations and related ecological transformations can be attained.
- This study offered valuable insights into the persistent ecological changes and their impacts on ancient communities in the dry western area of Egypt. The significance of interdisciplinary methodologies is underscored, which involves the integration of geomorphological inquiries and examination.
- The present study provided novel perspectives on paleoclimate variability in the Kharga Oasis during the mid-Holocene epoch. It underscores the importance of rock art records, including portrayals of prehistoric fauna and topography, in the investigation of paleoclimate change.

**Author Contributions:** H.I., W.A., H.G. and A.M.E.K. have made a substantial, direct, and intellectual contribution to the work and approved it for publication. All authors have read and agreed to the published version of the manuscript.

**Funding:** The research was funded by the British University in Egypt, project No. MFK21AH-2. The financial support from this institute played a crucial role in conducting the research and achieving its objectives.

**Data Availability Statement:** The datasets generated and/or analyzed during this study are available from the corresponding author upon reasonable request.

**Acknowledgments:** The authors would like to express their deep thanks to Mohamed Abdel Motamed and Hamouda Abdel Ghaffar for their sincere cooperation in conducting field studies in Kharga Oasis. The authors would also like to thank Ahmed Farrag, the guide of the Egyptian Ministry of Antiquities in the Kharga archaeological area, for his dedication and sincerity work with the project team.

**Conflicts of Interest:** The authors declare no conflict of interest. The funders had no role in the design of the study; in the collection, analyses, or interpretation of data; in the writing of the manuscript; or in the decision to publish the results.

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
