# Peer review of "Echoes of the Past: Unveiling the Kharga Oasis’ Cultural Heritage and Climate Vulnerability through Millennia"

_heritage, doi:10.3390/heritage6090335_

Round 1

Reviewer 1 Report

SUMMARY

The paper aims to provide fresh insights into the historical past of El-Kharga Oasis by examining both paleoclimatic conditions and rock art records. It underscores the importance of rock art as a valuable tool for studying changes in paleoclimate and their broader implications. The research confirms shifts in El-Kharga Oasis' climate, transitioning from periods of lush forests, abundant moisture, and domesticated animals to drier conditions and desertification.

COMMENTS

The paper's significance to the journal lies in its analysis of pictorial depictions on rock surfaces to investigate the ancient Egyptian society's susceptibility to climate variability. However, there is an opportunity for enhancement, particularly in terms of presentation, specifically in clearly addressing the research questions. While the title suggests a focus on climate vulnerability, the study predominantly delves into how paleoclimatic conditions influenced human migration.

To improve the paper, I propose the following revisions to the authors:

1. Articulate the research questions more precisely to outline the study's objectives and establish a more organized approach. Provide explicit answers to each research question. For instance, the fourth research question "What are the risks associated with paleoclimate change to the ancient Egyptian civilization in El-Kharga Oasis?" lacks a distinct response in the Results and Discussion section.

2. Discuss existing methods and solutions proposed for the digital restoration and safeguarding of petroglyphs. Specifically, investigate visual analytics approaches such as those detailed in references 10.1109/TVCG.2013.219 and 10.1145/2254556.2254658. Additionally, consider discussing projects like 3D Pitoti and IndianaMas, which have already conducted petroglyph analyses to study prehistoric populations.

3. Envisage future directions based on the attained results.

By incorporating these suggestions, the paper can be strengthened, providing a clearer focus on the research questions, acknowledging existing projects, and presenting prospects for the future.

Furthermore, the Conclusion section should be numbered as section 5.

The quality of English is satisfactory.

Reviewer 2 Report

The Sahara formation and expansion dominate the water cycle and climate system across North African, the Mediterranean and West Asian regions. The impact is worsening continuously even today and in the future. Any study in this area should be welcomed and encouraged. This paper makes a significant contribution in presenting the petroglyphs and rock arts from the El-Kharga Oasis region and analyzing their relationship with the climate change during the last 12,000 years. 

Improvements: 

1. The Abstract and Introduction state that the prehistoric migration from Kharga to Nile Valley started around 12,000 years ago was due to the northward shift of ITCZ, which caused a change in Egypt's savannah forests from abundant vegetation to an extremely dry desert. This seems in contradiction with the ITCZ mechanism and it's effect. Generally, when ITCZ shifts to north, it brings monsoon rains with it. Therefore, the northward shift 12,000 years ago made the Sahara green, not drier. 

2. Petroglyphs are great evidence reflecting climate and environment changes. The challenge is how to confidently estimate the times of their creations. More discussions may be added on how to determine the time periods when the rock arts were generated. 

The English is fine. Only minor editing is needed. 

Reviewer 3 Report

I found this an interesting and engaging manuscript to read.  I have a few suggestions to help strengthen the paper.  In no particular order these are:

1) BC vs. BP, the paper jumps between these, stick either with BCE or years before present.

2) Figure captions need far more detail, especially the multiframe figures such as 5.  This will also help clarify some of the text.

3) In a couple places the reasoning was a little unclear.  Line 353 in particular.  Here it was not clear to me how or why the art work implied seasonal migrations.  It may be that addressing comment 2 will also help clarify this.

4) Several of the figures cite other sources, but some of these are missing from the reference list.  I noted Wanner, Brookfield, and Vermeersch as ones mentioned in figures but not in the reference list. There may be others. 

5) There is some repetition in the paper, this maybe could be addressed along with some moderate editing for language. 

Overall the paper reads well, there is some repetition that could be cleaned up, and moderate editing for language would be helpful though not essential for understanding. 

Round 2

Reviewer 1 Report

The majority of my comments have been addressed by the authors.

I have two final points to make:

1) In your response to the review letter, you mentioned, "Based on the suggestions provided, we have conducted a comprehensive examination of traditional Petroglyph methods, the primary challenges related to this type of assessment in Egypt, and an exploration of the most advanced techniques in this context (such as visual analytics methodologies) as outlined in the cited references (DOI: 10.1109/TVCG.2013.219 and DOI: 10.1145/2254556.2254658)."

However, I noticed that the second reference was missing. I recommend adding the reference at the end of this sentence: "The integration of technological capabilities and visual discernment in the field of petroglyph research offers a valuable perspective known as visual analytics."

2) I would like to suggest a thorough review of all the references as there seem to be numerous errors, such as the reversal of first names and last names, duplicate references (e.g., references 70 and 71), missing page numbers, and other similar issues.

The quality of English is good.
